# Using a Process Approach to Pandemic Planning: A Case Study

Hana Tomaskova [1,*] and Erfan Babaee Tirkolaee [2]

1   Faculty of Informatics and Management, University of Hradec Kralove,
    50003 Hradec Kralove, Czech Republic
2   Department of Industrial Engineering, Istinye University, Istanbul 34010, Turkey; erfan.babaee@istinye.edu.tr
*   Correspondence: hana.tomaskova@uhk.cz

**Abstract:** The purpose of this article was to demonstrate the difference between a pandemic plan's textual prescription and its effective processing using graphical notation. Before creating a case study of the Business Process Model and Notation (BPMN) of the Czech Republic's pandemic plan, we conducted a systematic review of the process approach in pandemic planning and a document analysis of relevant public documents. The authors emphasized the opacity of hundreds of pages of text records in an explanatory case study and demonstrated the effectiveness of the process approach in reengineering and improving the response to such a critical situation. A potential extension to the automation and involvement of SMART technologies or process optimization through process mining techniques is presented as a future research topic.

**Keywords:** pandemic plan; business process model and notation; BPMN; process approach; a case study

## 1. Introduction

It is difficult to define the term "pandemic". Many different diseases have been identified as pandemics and have taken very different paths [1]. Morens, Folkers, and Fauci identified key features that can be identified for most pandemics in their publication [1]. Wide geographic spread, disease movement, high attack rates and explosiveness, low population immunity, novelty, infectiousness, contagiousness, and severity are among them. Grenan [2], on the other hand, defined the pandemic as a global epidemic. No matter how it is defined, pandemics have plagued our world almost since its beginnings, whether it was typhus, plague, cholera, flu, or the current SARS-CoV-2 pandemic. A pandemic is extremely dangerous because of its unpredictability, and it creates a crisis for everyone, not just those who are infected. As a result, pandemic planning is a critical regulation that represents one of the highest levels of crisis management. Shearer, Moss, McVernon, Ross, and McCaw [3] stated that planning is critical to mitigating the sudden and potentially catastrophic impact of an infectious disease pandemic on society. National pandemic policy documents cover a wide range of control options, frequently with broad recommendations for action. They also highlighted the gap between current analytical methods for decision making and their incorporation into these key plans. We mention the following publications devoted to the optimization, analysis, or decision support of processes or plans; dynamic simulation [4–9]; strategic management [10–16]; economic analyses; or information technology [17–20].

Automation, system integration, or an overarching process approach are all aimed at lowering costs while improving service quality, which is critical, not only for pandemic planning. As the name implies, the process approach focuses on the process and its activities. The process's outcomes should be definable and predictable. Another essential feature is that the process follows a logical and linear sequence or is functionally dependent on the procedures and resources that occur within the process. Some authors attempt to provide a solution for process model analysis. Melao and Pidd [21] and Tomaskova [22], for example, discussed the advantages and disadvantages of various modeling approaches used in

business process transformation. Glassey [23] compared three case study process modeling processes. Sadiq and Orlowska [24] employed graph reduction techniques to analyze process models. Other authors, such as van der Aalst, Reijer, Weijters et all [25] or Krogstie, Sindre and Jorgense [26], employed specialized tools, frameworks, and methodologies for process analysis and modeling.

The secondary goal was to conduct a literature review focusing on a process approach in pandemic planning publications. The primary goal of this article was to analyze the available pandemic documents and prepare an exploratory case study for the Czech Republic's pandemic plan using a process approach, specifically the Business Process Model and Notation (BPMN).

The remainder of the paper is structured as follows. The Czech healthcare system and crisis preparedness plan documents are briefly introduced as part of the basic information. The basic methods for this article are described in the Methods section. Then, a systematic review, a case study, document analysis, and BPMN follow. The main section contains the findings of a systematic review as well as model diagrams from an exploratory case study of the Czech pandemic plan in BPMN. The Discussion section discusses the paper's limitations as well as the gaps and inconsistencies in the current situation. The final section, Conclusion, summarizes the individual results and brings the paper to a close.

## 2. Background

The Background section is dedicated to introducing key facts. There are separate sections devoted to the Czech health system and hospital crisis preparedness documents.

### 2.1. Czech Health System

According to Alexa et al. [27], the Czech Republic has a statutory health insurance (SHI) system based on compulsory membership in a health insurance fund, seven of which existed in 2014. The funds are self-governing, quasi-public entities that act as both payers and purchasers of care. The Czech Republic's core health legislation was enacted in the 1990s and has only been slightly modified since then. Taxation is the foundation of financing. These taxes are taken out of employee or self-employment earnings. There is also the possibility that the healthcare system will rely solely on these funds, especially for those who are unable to pay the fee themselves (children, students, pensioners, or temporarily unemployed people).

Fall and Glocker [28] evaluate the Czech healthcare system so that it performs well in terms of health outcomes when compared to other Central East European economies that inherited similar health systems after the transition and have been converging to OECD averages.

### 2.2. Crisis Preparedness Documents of Hospitals

There are numerous ways to ensure that the hospital runs smoothly, including rule compliance, the development of an effective crisis plan and its stages, or state aid. Crisis documentation is the last thing that can help hospitals. A set of documents used for an organization's crisis management is known as crisis documentation. This is also part of the organization's management documentation, and many people work on it. These individuals are responsible for compiling this documentation, and if necessary, participating in its implementation in practice [29]. The documentation includes four plans: a crisis preparedness plan, a trauma plan, a pandemic plan, and a crisis plan.

#### 2.2.1. Crisis Plan

The crisis preparedness plan is designed for the natural and legal persons who create it, and it serves as their primary planning document. This plan is used both inside and outside the organization, i.e., among other entities. The plan's goal is to safeguard people's health and property. It is made up of primary, auxiliary, and operational components [30].

### 2.2.2. Crisis Preparedness Plan

The crisis preparedness plan is designed for the natural and legal persons who create it, and it serves as their primary planning document. This plan is used both inside and outside the organization, i.e., among other entities. The plan's goal is to safeguard people's health and property. It is made up of primary, auxiliary, and operational components [31].

### 2.2.3. Traumatology Plan

This plan is used by the health system to ensure that the hospital runs smoothly and that the crisis plan tasks are completed. It includes instructions for dealing with a variety of injuries (first aid, burns, electric shock, etc.). It contains contact information for various medical facilities as well as for medical staff, in addition to information on first aid kits and their contents. It is made up of three parts: primary, auxiliary, and operational [32].

### 2.2.4. Pandemic Plan

Pandemics, whether mild, moderate, or severe, affect a large portion of the population and necessitate a rapid and effective response for months or even years. As a result, countries are developing plans outlining their strategic response to a pandemic, which will be supplemented by operational plans at the national level. Pandemic preparedness is thus a continuous process of planning, implementing, reviewing, and putting national and transnational pandemic preparedness plans into action. This plan must be updated whenever global guidelines or evidence bases change, as well as any exercise or change in national or international legislation pertaining to infectious disease prevention and control occur [33]. Figure 1 depicts the cycle of key elements of pandemic preparedness.

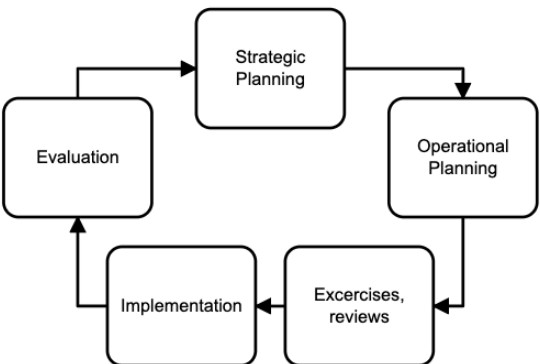

**Figure 1.** Key elements of pandemic preparedness.

## 3. Methods

The main techniques used in this article are described in the Methods section. The four fundamental methods presented here are systematic review, case study, document analysis, and BPMN.

### 3.1. Systematic Review

It is difficult to find a systematic literature review methodology for multidisciplinary or IT areas. As a result, we used Kitchenham's article [34], which states that a systematic review of the literature for IT should include three basic things. The first step is to identify the research question or research goal for the entire study. Following that is an equally important organization of the impartial and extensive analysis of related publications, and finally, the establishment of explicit inclusion and exclusion criteria.

We developed a research aim to examine the current level of use of the process approach in pandemic planning. The analysis procedure and criteria are detailed in the sections that follow.

### 3.1.1. Eligibility Criteria

The primary sample of the study is comprised of the publications listed in the two largest databases, Web of Science (WOS) of Clarivate Analytics and Scopus, that contain the search strings, regardless of the date of publication.

We established the following inclusion criteria (IC) and exclusion criterion (EC) as conditions for including/excluding publications from the final review:

Inclusion criteria and exclusion criterion are:

- IC1: The publication is concerned with the pandemic plan.
- IC2: The publication includes a model.
- IC3: A modeling or simulation method is used in the publication results.
- EC1:The text of the article is written in a language other than English.

### 3.1.2. Information Sources and Search

The WOS and Scopus databases were chosen as the primary data sources for the study. These databases contain publications that have been subjected to a review process, which is regarded as fundamental in scientific circles, and the outputs provide complete information suitable for analysis. We ran an advanced search for the search terms listed below. WOS search strings were:

$$AB = (``Pandemic\ plan"\ AND\ (process)), \qquad (1)$$

and Scopus search strings were:

$$TITLE - ABS - KEY(``Pandemic\ plan"\ AND\ (process)). \qquad (2)$$

### 3.1.3. Study Selection

We divided the publication screening process into several stages. In the first phase, we evaluated the document's title and abstract using the aforementioned exclusion criteria. During full-text reading, we considered the remaining publications and included two independent evaluators who verified our findings. We made no exclusions based on methodological quality.

We looked at the studies that went through the introductory network from a variety of perspectives and coded them according to various criteria.

The review's limitations restrict the analysis to English language publications published in the WOS and Scopus databases on 15 March 2021, and containing specific search keywords. This limitation may have resulted in the omission of some relevant studies published in other languages after 15 March 2021, or the inability to link a search query to a document.

### 3.1.4. Data Collection Process

For data collection, an advanced search in the WOS and Scopus databases was used. For each search string, the appropriate number of search queries were compiled. The documents were then examined to see if they did not meet the EC1 criterion and but did meet the IC1–IC3 Criteria, using the Kitchenham [34] procedure and the Preferred Reporting Items for Systematic Controls and Meta-Analyses (PRISMA) [35].

The following equation was used to determine whether a publication was a "final output publication (FOP)":

$$FOP = IC1\ AND\ (IC2\ OR\ IC3)\ NOT\ EC1 \qquad (3)$$

In cases of doubt or inconsistency, the principal author of this publication made the final decision. Figure 2 shows a thematically adapted PRISMA flowchart with detailed phases, including quantifiers.

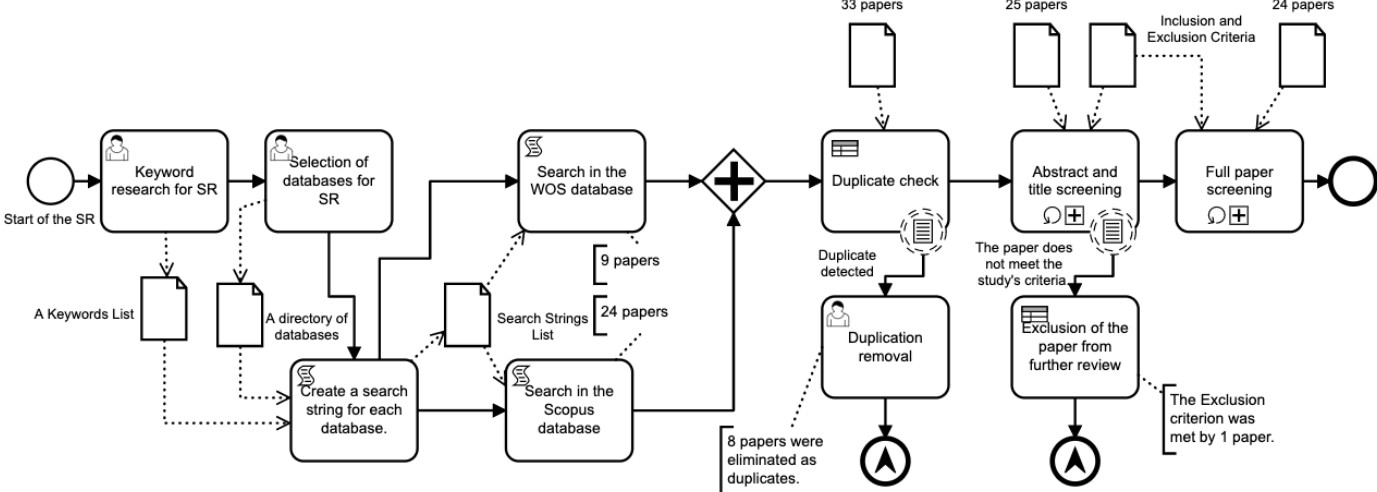

**Figure 2.** Process of Systematic Review.

### 3.2. Case Study

A case study, as defined by Gerring [36], is "an intensive study of a single unit for the purpose of understanding a larger class of (similar) units". The case study method, according to Holzer [37], can employ both quantitative and qualitative evidence. Yin [38] added that this information can come from observations, verbal records, and fieldwork, as well as a variety of data collection methods such as ethnography and participant observation.

There are several subdivisions within the more generalized category of case studies, each of which is custom selected for use based on the investigator's goals and/or objectives. According to Ogawa and Malen [39], the exploratory case study aims to broaden the understanding of complex social phenomena. It was used as the first step when extensive empirical research on the topic of interest has not yet been conducted. When the topic is unfamiliar or stereotypical views are imposed, this approach may be justified. It is possible to better define a problem, propose hypotheses to be tested later, generate ideas for new services, and gather feedback on a new concept.

### 3.3. Document Analysis

According to Corbin and Strauss [40] or Rapley [41], document analysis is a systematic procedure for checking or evaluating documents—both printed and electronic materials. Document analysis, like other qualitative research methods, necessitates the examination and interpretation of data to gain meaning, understanding, and empirical knowledge.

### 3.4. Business Process Model and Notation

In general, the BPMN can be thought of as a language for developing business process models or as a standard for modeling business processes. This notation was made completely open by the Business Process Management Initiative (BPMI). The BPMN may appear to be similar to flowcharts or Petri nets, but it actually provides far more sophisticated tools for describing and simulating behavior, as well as greater user friendliness as, for example, Silver [42] or Nisler and Tomáková [43] point out. In 2004, the first version of the BPMI, BPMN1.0, WAS released. The BPMI merged with the Object Management Group (OMG) in 2005. The BPMN specification document was published by the OMG the following year. BPMN2.0 was developed by the OMG in 2010, and the current version, BPMN2.0.2, was released in December 2013. The evolution of BPMN and its notation is a frequent topic in BPMN publications; we can mention, for example, the works of authors; Kocbek et al. [44], Chinosi and Trombetta [45], White [46], Tomášková and Weber [47], Van der Aalst et al. [48] or Recker [49]. This notation's primary goal is to be used in a variety of processes. This is understandable for non-specialists while also allowing for sharing and discussion among experts from various fields, as stated by OMG [50].

All users of BPMN 2.0 have access to four simple classes of graphical elements. Flow objects, connecting objects, swimming lanes, and artifacts are examples of these. The sections of the diagram that make up the overall workflow are called flow objects. Events, activities, and gateways are the three primary flow objects. The connection objects' role is to link flow objects together. As connecting objects, we may include sequential flow, message flow, and association. Swimming pools are represented as rectangular frames in BPMN as participants in the business process. Individual segments of the pool can be divided by lanes in the swimming pool. Swimming pools can be built vertically or horizontally. The process runs from left to right for horizontal swimming lanes and from top to bottom for vertical swimming planes.

## 4. Results

The results section is divided into two sections: the findings of a systematic review and a brief meta-analysis of the included papers, and diagrams with descriptions of the exploratory case study of the relevant part of the pandemic plan based on the documentary analysis of the relevant documents. A pandemic plan is a trauma plan that applies to a wide range of locations. It also handles much more serious crises than the trauma plan. This also corresponds to its three parts: primary, operational, and partly auxiliary.

### 4.1. the Findings of a Systematic Review

We analyzed 33 papers, of which eight were duplicates and one met the EC1 criterion. All included 24 publications were released between 1994 and 2021; 21 were articles, one was a book, and two were reviews. Only four publications had a single author. If we look at the year of issue, $Q_0$ (minimum) = 1994, $Q_4$ (maximum) = 2021, $Q_1$ = 2005 and $Q_3$ = 2015.

Regarding publications, pandemic planning in relation to viral diseases such as influenza or the A/H1N1 virus was found to be optimal. Publications addressing the issue of coronavirus SARS in pandemic planning became popular in both 2005 and 2016. Since 2020, the new coronavirus COVID-19 has urged publications to focus on pandemic planning. Figure 3 depicts the thematic division of publications.

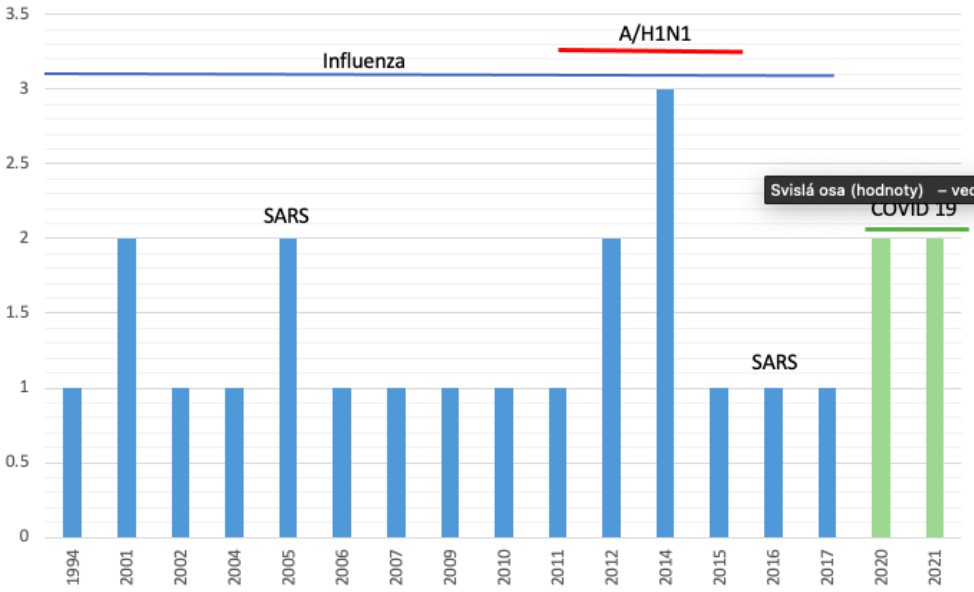

**Figure 3.** Year of issue and topic.

Table 1 provides an overview of how the individual publications met the IC. Publications that met IC1 were fully committed to pandemic plans but did not address their implementation process. They instead addressed how states should prepare, analyzed previous pandemic data, addressed the use of a pandemic plan in remote, provincial, or abo-

riginal areas, or managed the allocation and supply of medicines, vaccines, and devices in pandemic plans.

**Table 1.** The findings of a systematic review.

| Publications | IC1 | IC2 | IC3 | FOP |
|---|---|---|---|---|
| [51–68] | ✓ | | | × |
| [69,70] | | ✓ | ✓ | × |
| [71] | | ✓ | | × |
| [72–74] | | | ✓ | × |

4.1.1. A Brief Overview of the Papers Included

Tamblyn [61] discussed pandemic preparedness in Canada. This focused on the need to update the pandemic plan's objectives and roles by the influence of the present improved influenza vaccine production capacity.

Paget and Aguilera [72] conducted an EU-wide questionnaire survey. They focused on flu surveillance methods. They stated that all countries, with the exception of one, had or were preparing a pandemic plan. The coordination of these various national plans at the European level would most likely contribute to their increased impact and effectiveness.

According to Gensheimer et al. [58,62], the United States is prepared for influenza pandemics. They concentrated on coordinating planning at the local, national, and state levels.

Jennings and Lush [59] addressed New Zealand's pandemic preparedness, drawing conclusions based on a monthly pandemic training simulation.

According to Kort, Sturart, and Bonotovics [55], the SARS crisis exposed critical gaps in Ontario's ability to respond to health emergencies and highlighted the need for better emergency response planning.

Ward and colleagues [73] discussed the consequences of avian influenza cross-transmission and proposed updating pandemic plans to optimize the use of antiviral drugs. They emphasize the importance of addressing processes for collecting and reporting data from treated patients.

Itzwerth et al. [74] highlighted hospitals' vulnerability during the spread of an influenza pandemic. The critical relationship that exists between the health sector and other sectors is poorly understood and addressed. They made the point that hospitals rely on a critical infrastructure that exists outside of the organization, and that existing plans do not adequately account for the complexity and interdependence of the systems on which hospitals rely.

Straetemans et al. [63] concentrated on strategies for determining vaccine resentments during the early stages of an influenza pandemic.

Nishiura, Wilson, and Bakerí [71] investigated the efficacy of quarantine as a border control measure. The effectiveness of quarantine at island borders was modeled as a relative reduction in the risk of releasing infectious individuals into the community by analyzing the detailed epidemiological characteristics of influenza.

Zoutman et al. [67] designed and analyzed a questionnaire survey of acute care hospitals in Ontario to assess their pandemic influenza preparedness plans.

Charania and Tsuji [51,66] designed and evaluated a semi-structured questionnaire survey on the barriers to responding to the 2009 H1N1 pandemic and future improvements.

Cousineau and colleagues [56] examined Quebec legislation and proposed the precautionary principle as a possible way to justify the state's access to genomic databases and their use for research purposes.

Cauchemez and colleagues [65] focused on school dropouts during an influenza pandemic and presented the diverse experiences of twelve countries (Bulgaria, China, France, Hong Kong Special Administrative Region (SAR), Italy, Japan, New Zealand, Serbia, South Africa, Thailand, the United Kingdom, and the United States).

To describe the relationships between immunization programs and emergency preparedness programs, Seib et al. [57] conducted a survey among immunization program (IPM) managers.

Charania and Tsuji [64] conducted a healthcare survey in three remote and isolated Canadian communities of sub-Arctic Ontario's first nations. They concentrated on the various risks that such communities face.

Miller et al. [68] described the application and efficacy of the participatory action (PAR) research framework to better understand the public perceptions of pandemic influenza risks. Because the H1N1 influenza pandemic in Oceania and the Americas affected indigenous peoples more than non-indigenous peoples, the authors focused on ensuring that Australian national, state, and territorial pandemic plans adequately reflected the at-risk status of indigenous and Torres Strait Islander peoples and promoted meaningful cooperation with communities in order to mitigate this risk.

According to Medina [60], pandemics of influenza A occur at 10- to 50-year intervals. As a result, the author sought to correct the oversight of pandemic preparedness plans by providing a summary of guidelines and recommendations from international health organizations, pandemic death experts, and experienced mass death management experts.

Sambala and Manderson [52] prepared and analyzed a questionnaire survey with government policymakers and people working at the political level in various NGOs to assess Malawi's level of preparedness and how it translated its national influenza plan into response.

Kirlin [53] focused on the question "How did previous pandemic planning shape COVID-19 responses?" This points to insufficient data warehouse systems and serious economic and social dislocations.

In addition to the well-known exponential range, Öz [69] introduced a model that is structurally capable of describing both government actions and individual reactions. In addition, the weather effect is included. This method demonstrates a quantitative method for tracking these dynamic effects. This allows one to calculate the impact of various private or public measures put in place to combat a pandemic at time t.

Pendharkar and colleagues [54] described The Medical Emergency-Pandemic Operations Command (MEOC), a pandemic plan that was designed and implemented from March to May 2020 and then re-escalated in October 2020.

Koç and Türkoğlu [70] concentrated on artificial intelligence-based models that play an important role in predicting the demand for medical equipment during infectious disease outbreaks. In the experimental studies, 77-day COVID-19 data from Turkey's statistics were used. The proposed system was tested using data from the last 9 days of this dataset, and its performance was calculated using the MAPE and R2 statistical algorithms.

### 4.1.2. Papers Meeting IC2 and IC3 Criteria

The publications listed below proposed a model or the use of simulation methods.

A novel hybrid scheme for forecasting the demand for medical equipment and outbreak spread of COVID-19 was presented by Koç and Türkoğlu [70]. The proposed model employs a multi-layer long short-term meemory (LSTM) network to accurately predict the number of respiratory equipment and intensive care beds required in the event of a COVID-19 pandemic.

To reduce the impact of a pandemic, Itzwerth et al. [74] recommended that hospital management use business continuity management and business continuity planning to focus on risk analysis and the potential effects of such threats.

Öz [69] built an SEIR-based model that includes the population's risk perception via an additional differential equation and employs an implicit time-dependent transmission rate.

All publications agree on the importance of having an up-to-date and comprehensive pandemic plan. Nonetheless, none of them address its design as a process that can be optimized or subjected to security audits.

### 4.2. BPMN Compilation of a Primary Component of a Pandemic Plan

In the first step, vital information about healthcare providers must be gathered. Subsequently, the activity that the given subject (hospital) would perform in the event of a pandemic is defined. Moreover, as with the trauma plan, all possible external sources of risks and threats that could lead to mass health damage must be identified with the assistance of the Fire and Rescue Service. Even here, all internal risks that could jeopardize the hospital's operation in a crisis like a pandemic were identified. The provider must also characterize all types of different injuries and illnesses for which a given pandemic plan is compiled, as well as define the measures that must be implemented when fulfilling a pandemic plan. This step will be carried out based on both external and internal risk analysis, as well as the disabilities for which the pandemic plan is intended. Figure 4 represents the corresponding diagram of the primary part model.

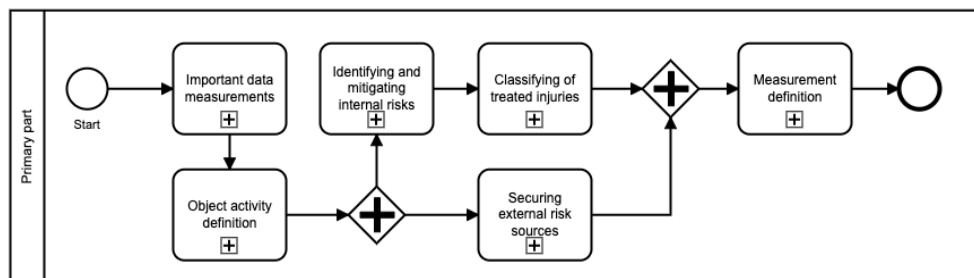

**Figure 4.** Primary part of pandemic plan.

### 4.2.1. Important Data Measurements

The pandemic plan affects a much larger population. As a result, certain activities are decided by the pandemic group rather than the hospital itself. In our case, we will refer to this entity as the "pandemic group" in general.

The requirement of critical data by healthcare providers in the hospital identifies the part of the staff that makes decisions about the various processes that occur in the hospital. In our case, it is the hospital administration.

Therefore, the pandemic group first drafts the requests, and sends them to the hospital. Their leadership captures the message here. This, in turn, ensures that all data from all members of the management are secure. Only personal data such as a person's name, surname, phone number, and residential address will be collected in the case of a natural person. In the case of a legal entity, the address and name of the company will be determined in addition to personal data. This cycle is repeated until all management members have died. The hospital identification number is then assigned. This code serves the same function as a descriptive number. Finally, other crucial information is added, and the entire file is returned to the pandemic group. It receives the message, processes it, and stores it in its database. The entire diagram can be seen in Figure 5.

### 4.2.2. Object Activity Definition

As with the previous part of the pandemic plan, a pandemic group, in addition to hospital management, will make this decision. The pandemic group's leadership will meet first. This will then result in the creation of a report inquiring about the subject's activities. This message is then forwarded to the hospital. The report is captured by the hospital's management, who then searches the database for basic information about their facility. After that, the message is forwarded to the pandemic group. This report will then define the hospital's activities and inform the hospital of this decision. It only saves the message's outcome in the database. A pandemic group will also store the result of the definition of the object's activity in its database. Figure 6 depicts this process in considerable detail.

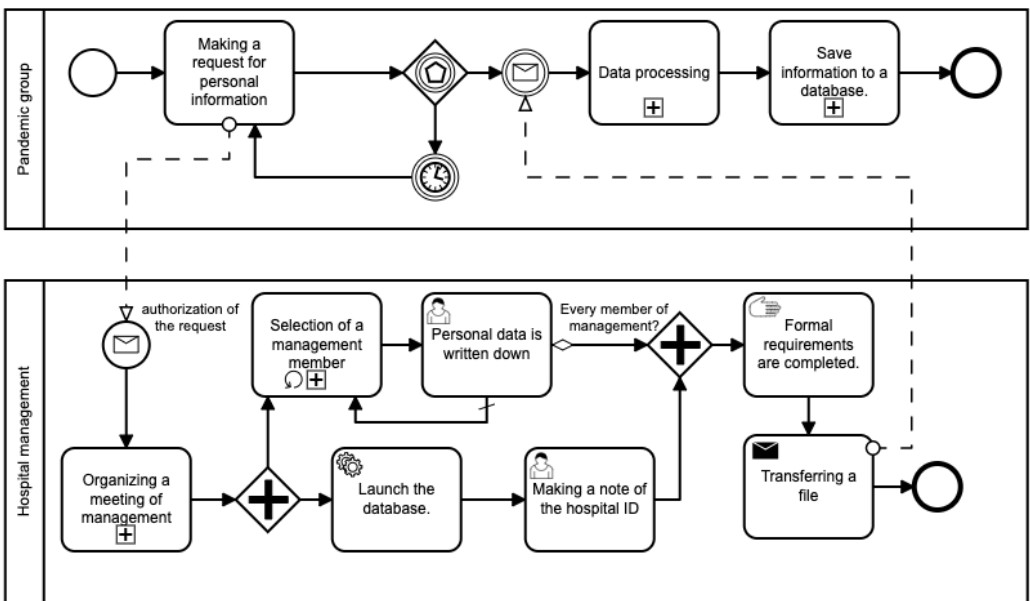

**Figure 5.** Important data measurements.

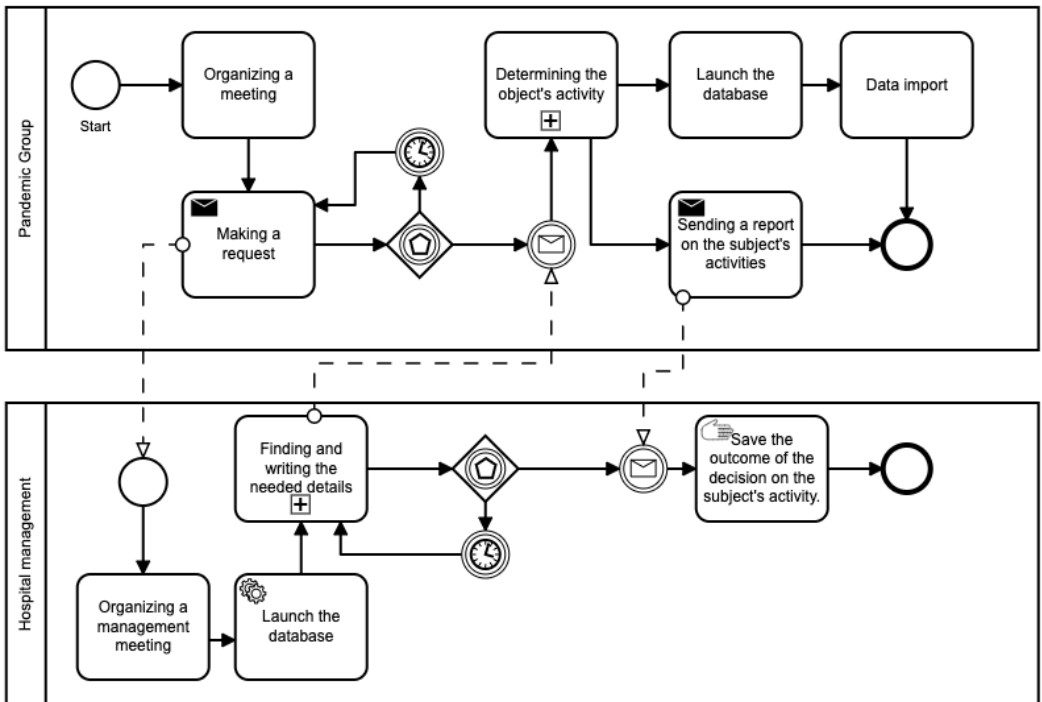

**Figure 6.** Object activity definition.

### 4.3. BPMN Compilation of a Operational Part of a Pandemic Plan

The operational section is primarily concerned with what will be done during the event—that is, whether a pandemic plan is implemented. First and foremost, the medical facility must ensure that it has enough protective equipment and disinfectants to treat patients. They must also ensure that they have an adequate number of paramedics. Furthermore, specific procedures and clinics where patients will be treated are established. As a result, so-called re- profiling, i.e., inpatient care, must be performed at all times. Figure 7 depicts the operational component of the pandemic plan.

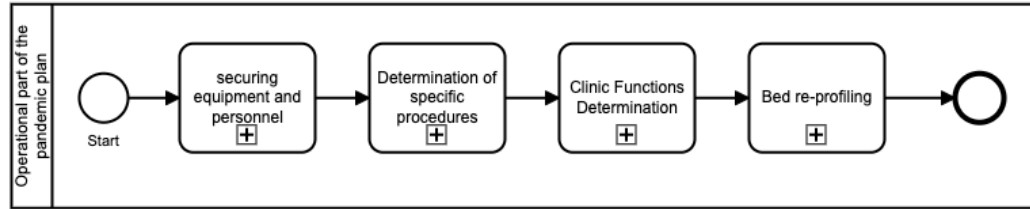

**Figure 7.** Operational part.

### 4.3.1. Clinic Function Determination

The hospital administration and the pandemic group work together to determine the functions of individual clinics. The pandemic group is formed first, and then all potential threats that led to the pandemic plan are documented. They then send this message to the hospital, where management takes over. This, in turn, has two responsibilities. The first step is to process the received files. The second task is for the hospital administration to list the current functions of the clinics, as well as other data, using a database. Following that, they will adjust the functions of all their clinics by completing the two previous tasks. The outcome is eventually returned to the pandemic group. They interpret the message and save it in the database. The hospital administration will do the same. This result is then saved in both lines' databases. Figure 8 depicts the diagram.

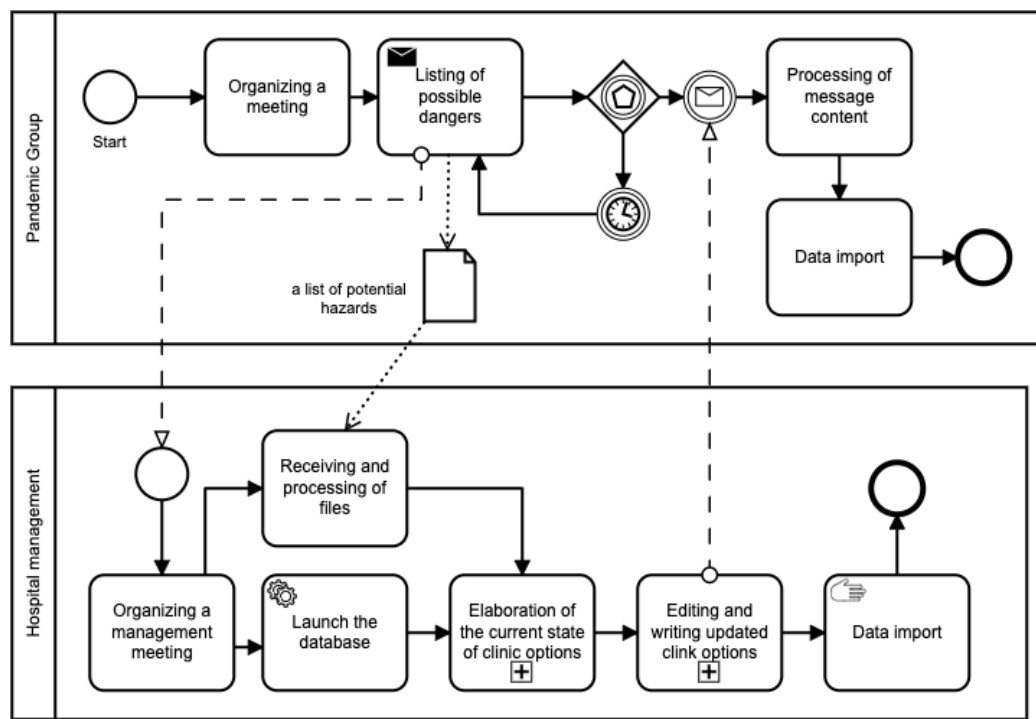

**Figure 8.** Clinic function determination.

### 4.3.2. Bed Re-Profiling

Bed re-profiling is a term that includes an assessment of the patient's health, a count of the total number of patients, and a count of the number of health professionals. If any of these data points were unsatisfactory (too few healthcare professionals or too many patients), the issue would have to be resolved quickly. For example, using the army's assistance or transporting patients to other hospitals. Logistical security must also be provided during bed re-profiling. It ensures that there is always enough food, medicine, and other supplies in the hospital. The entire process is depicted in detail in Figure 9 below.

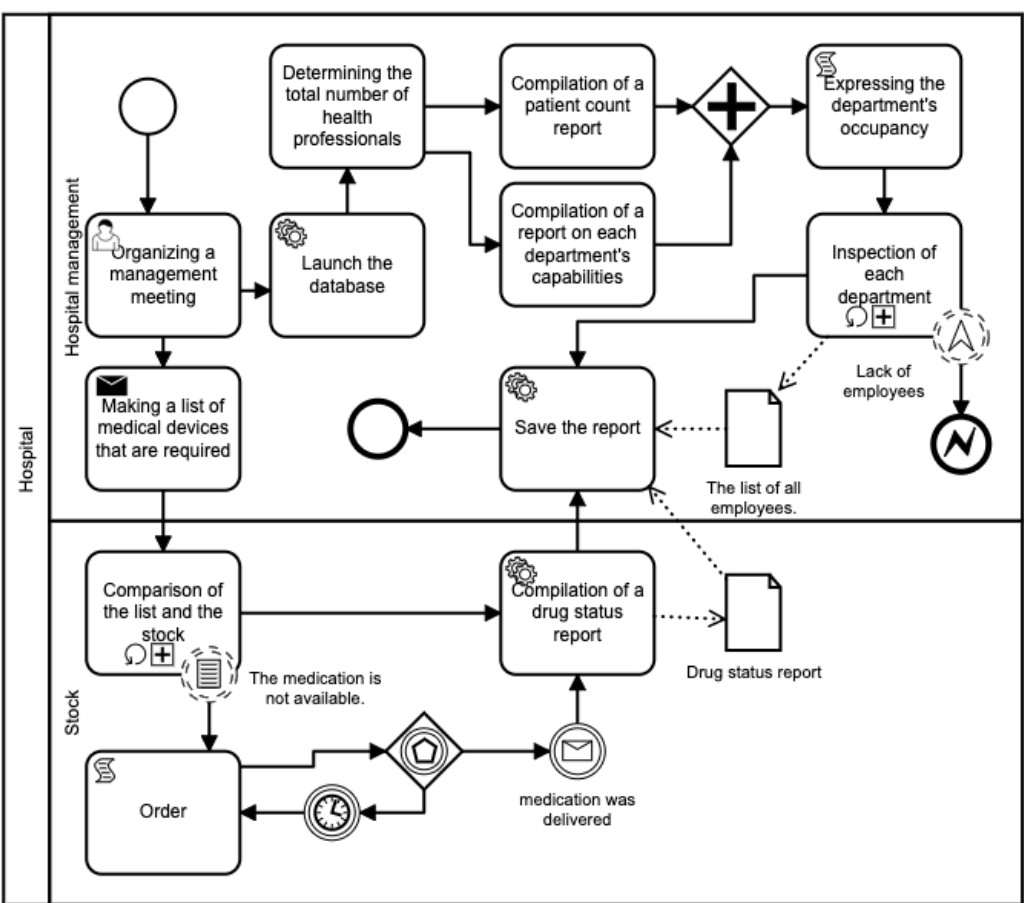

**Figure 9.** Bed re-profiling.

The hospital administration will be called first. Then, it has two duties. The first is to determine the total number of health professionals, each department's capacity, and the total number of currently ill patients using a database. They then calculates the department's percentage that is occupied based on this information and determine whether there are enough health professionals in each department. If a ward is discovered to have an insufficient number of health professionals, it will be escalated to certain entities. This could be the region's management or another hospital.

The hospital administration's next step is to determine whether they have enough medication. As a result, the hospital administration will create a list of medicines required and send it to the warehouse. They look through the list for medication. If none are available, they will be ordered. If the medication is in stock, it will be removed from the list. They check whether the requested medicines have arrived after going through the entire list in the warehouse. If the drugs have already been delivered, their names will be removed from the list. Following that, the drug status reports are returned to hospital management. They receive messages and save them to the database.

### 4.4. BPMN Compilation of an Auxiliary Part of a Pandemic Plan

We will only provide contact numbers for important people (patients, healthcare professionals, etc.) in the auxiliary segment, note the pandemic group members, write down all important information gathered during the pandemic, and list all consumed protective equipment and disinfectants. The final step is to ensure that all contracts that arose during the pandemic plan's implementation are written and paid. Figure 10 depicts the auxiliary component of the pandemic plan.

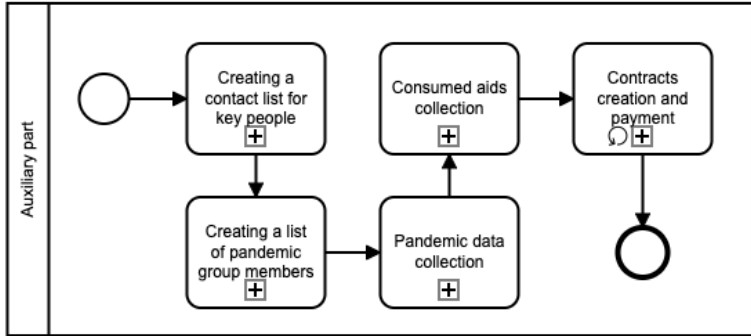

**Figure 10.** An auxiliary part of a pandemic plan.

### 4.4.1. Creating a List of Pandemic Group Members

During this stage of the process, the hospital administration is summoned, and a request for the disclosure of personal data to all members of the pandemic group is written. The hospital administration will then forward this message to the pandemic group. The report must receive and record all relevant information to each group member. The data are then returned to the hospital, where the report is taken over by management and the detected data are stored in the database. Figure 11 depicts the diagram graphically.

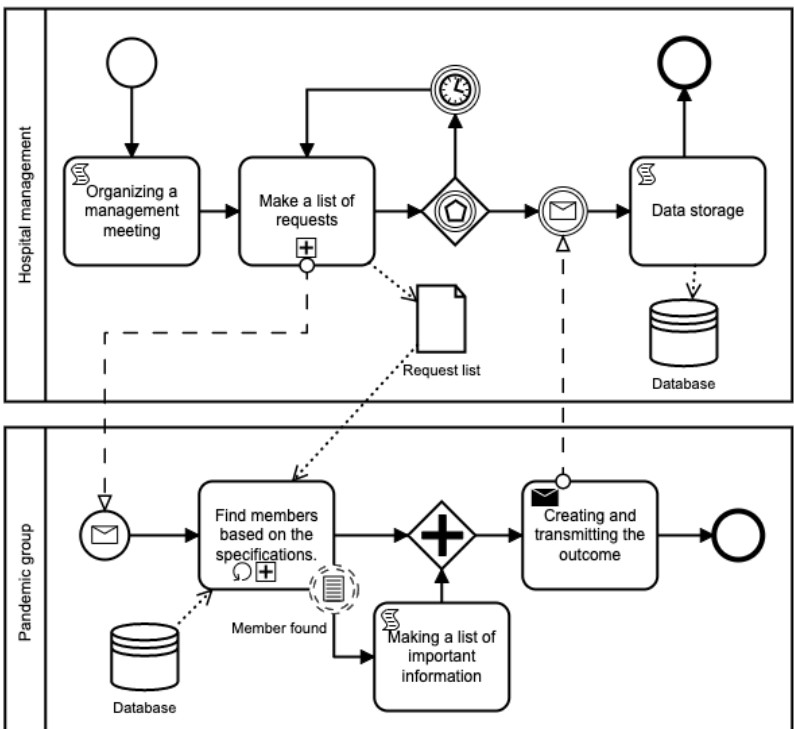

**Figure 11.** Creating a list of pandemic group members.

### 4.4.2. Collection of Consumed Aids

The final graph is a compilation of aids consumed during the implementation of the pandemic plan. Initially, the corresponding hospital employee writes a request regarding the medication consumed. This request is routed to the warehouse. Other hospital employees will inspect the entire warehouse and record the missing medical aids. After the inspection of the entire stock, the list of consumed aids is returned to the employee who writes the list of these aids in the database. Figure 12 depicts the process of counting the volume of consumed general aids.

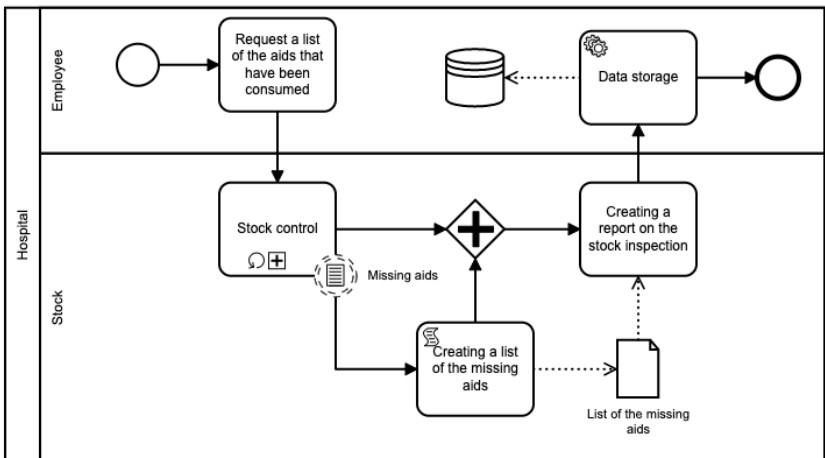

**Figure 12.** Collection of consumed aids.

## 5. Discussion

Pandemic planning research has confirmed a critical need for a comprehensive solution, as authors such as Tamblyn [61] , Gensheimer et al. [58,62], Kort, Sturart, and Bonotovics [55], Ward and colleagues [73], Itzwerth et al. [74], Cousineau and colleagues [56], Miller et al. [68], Medina [60] or Kirlin [53] have already stated in previous publications. The unexpected finding was that no one had yet addressed or built the pandemic plan as a process. Koç and Türkoğlu [70] forecasted using a multilayer long short-term memory (LSTM) network, Öz [69] developed an SEIR-based model, and Itzwerth et al. [74] suggested using business continuity management and business continuity planning. This research can help to fill that void. It is critical to holistically examine the problem and to connect the individual activities of various subjects.

An intriguing finding was the ongoing dissatisfaction with current pandemic plans across the world as well as the growing request for their solutions and learning from previous pandemics. The Czech Republic, as seen in our exploratory case study, also lacks a representative pandemic plan. A documentary analysis of available materials demonstrates the obsolescence of procedures that are not supported by modern technology and misses a comprehensive view of the problem. Unfortunately, pandemic planning is a "seasonal problem." When a pandemic occurs, there is insufficient time to analyze and optimize the procedures, and it is necessary to address and act. However, once the pandemic has ended, the consequences of the pandemic must be addressed, and pandemic planning is postponed.

In addition to the significant flaws of the current state, we can point to insufficient or even non-existent automation. Many activities rely on manually rewriting data from and to the database. In this case, we recommend using scripts sent through secure channels or smart devices to automate the process. A wide range of activities can be classified as acts of service. However, these service activities take up the time of important people who should be making key decisions at the time. Instead, they deal with the administrative tasks on the agenda. Automation and smart devices are also viable options. Most ERP systems should be automated; for example, Samaranayake [75] has presented a system of process integration for functional applications, the automation for business workflows, and additional functionalities for process optimization, which can also be used for pandemic planning, or as Martin-Navarro, Sancho, and Berro [76] describe the process of implementing information systems and management automation, with an emphasis on the ERP system. We can use the problem of patients and their predictions in the Czech Republic as a functional example of the applicability of modeling and process approaches. First, Marešová and others [77] presented the socio–economic aspects of Alzheimer's disease. Mohelská et al [78] compiled a case study of the costs of treatment of these patients. Cimler and others [4] analyzed suitable simulation models for population prediction, and finally,

Cimler et al [79] predicted Alzheimer's disease treatment and care costs in European countries using a simulation model. Simultaneously, Kopecky and Tomášková [80] linked costs and process approach, and Tomášková, Kopecký, and Marešová [81] compiled the Alzheimer's disease process cost management model.

Finally, the recommendation is to analyze the individual steps of the pandemic plan, compare them to the available data, and optimize the pandemic plans. Comparing the various methods or approaches used will allow for more effective adaptation to other pandemics in less time than is currently the case.

## 6. Conclusions

This article presented basic knowledge of the Czech healthcare system and crisis preparedness documents, as well as methods such as systematic review, document analysis, case study, or BPMN. The findings of the systematic review then pointed to the current research literature's narrow focus on using the process approach in pandemic planning. The document analysis and exploratory case study not only demonstrated the suitability and effectiveness of process modeling, but also revealed numerous structural errors or unnecessary activities that would be eliminated by basic process automation.

The main limitation of this paper is the limited availability or lack of detailed documents of pandemic plans. As a result, the study is at a higher level of abstraction, and further specificity of activities and responsible persons will be future research. Another limitation is the lack of research studies addressing a priori processes in pandemic planning. Although we attempted to approach the documentary analysis and subsequent exploratory case study objectively, one author's bias could be a negative limitation.

A process mining approach was mentioned in the introduction as a possible extension for optimizing process workflows. The entire process can be analyzed using throughput analysis, safety analysis, or an activity-based costing approach. Future research will require the analysis of specific data on real-world processes, which may differ significantly from the prescribed official plans.

Although pandemic planning is currently a top priority, decisions about its processes are predominantly made by bureaucrats. As a result, current processes lack an overview, the interconnection of different industries, or reflection on simple project procedures, such as delivery and orders, or the time of work on an activity that can occur concurrently with another key activity. As a result, any evaluation of the optimality of procedures or processes may improve the efficiency of future processes.

**Author Contributions:** Conceptualization, methodology, software, investigation, resources, data curation, writing—original draft preparation, writing—review and editing, visualization, H.T.; validation, formal analysis, E.B.T. and H.T.; supervision, E.B.T. All authors have read and agreed to the published version of the manuscript.

**Funding:** The research has been partially supported by the Faculty of Informatics and Management UHK specific research project 2107, Integration of Departmental Research Activities and Students' Research Activities Support.

**Institutional Review Board Statement:** Not applicable.

**Informed Consent Statement:** Not applicable.

**Data Availability Statement:** Not applicable.

**Acknowledgments:** The author would like to thank students Nosek K. and Novak J. for their cooperation on the topic.

**Conflicts of Interest:** The authors declare no conflict of interest.

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
