# Peer review of "Using a Process Approach to Pandemic Planning: A Case Study"

_applsci, doi:10.3390/app11094121_

Round 1

Reviewer 1 Report

The topic is really very interesting and relevant in the context of both pandemic and process management. However, the authors should follow the instructions of the template.

The main methods and the main results used must be described in the abstract (summarized). At the end of the abstract, future research should be summarized in a few sentences.

The introduction is weak. In my opinion you need to include more references. Some sentences need to be based on the references. For example, the first paragraph of the introduction. Who says that? It is also not very clear how certain sentences are related to the scientific sources cited. For instance, strategic management (30 row), etc. It is also suggested to refer to specific authors in the paper rather than to write The paper [20] compares, The article [21] employs, etc. (33, 34 rows). Rational need to be logical, based on the references. The main goal of the research should be described in the Introduction.

The second chapter Czech health system go over the Czech health care system, including a brief history of it. The same remark. Rational need to be logical, based on the references.  Who says that?

My second issue grounds on the research methodology. The authors should follow the instructions of the template. It should be the chapter Methods. I see that you are using another method, i.e. document analysis (this is Crisis preparedness documents of hospitals). This must also be described. Not just the systematic literature review.

Where is a Result Chapter?  The discussion section seems rather brief and not very detailed. I recommend providing more in-depth discussion in this section.

The Conclusion chapter is too short. The results should be summarized and linked to the introductory text. Also, recommend adding limitations in the paper.

Author Response

Thank you for your valuable suggestions; partial responses are included in the attached file.

Reviewer 2 Report

The authors of the article discussed an interesting topic of the process approach to pandemic planning. The issue is definitely timely in view of the failure of many governments to deal with the pandemic.
Unfortunately, in my opinion, the article must undergo significant modifications so that it meets the requirements for scientific articles. First of all, the authors were to present a case study of the Czech Republic - the case study method was not described, however, and the authors focused on the description of the literature review method. The case study itself was not presented either.
In the Introduction section: the Authors mention "other authors" - they do not mention them by name. Why in the sentence: "The process approach, automation, and system integration aim to reduce costs while improving service quality, which is critical in many areas" the authors separate these issues, since the process approach itself covers both "automation" and "system integration" ? The "Introduction" also lacks the explicit purpose of the article.

In the second part, there are no references to the literature when the Czech Republic Health System is described. 

Also "Discussion part" and "Conclusions" need to be strenghtened.

Provided Literature only in 20% includes the positions issued after 2016 year. There is no literature strictly related to the process approach. Some items need to be completed, e.g. 26-29. 

Author Response

(The authors gave the same response as above.)

Reviewer 3 Report

An interesting paper. Following a systematic review of prior literature, the author(s) construct a pandemic plan in BPMN, depicting the primary, operational, and auxiliary phases. The author(s) suggest that the paper fills a literature gap by addressing the Pandemic Plan as a comprehensive process.

A final search of 24 publications was made, but there is no mention of the number of articles observed (are they the references 42-62 in table 1? – although this table only refers to “Inclusion Criteria”, Line 145) or the time frame. There is some confusion concerning the terms “papers” and “publications”  in the various Figures. How many “Journals” were identified, and how many “articles” were viewed in each journal? The author(s) refer to “a case study”, but there is no mention in the text about where this case study was undertaken.

Both the discussion and conclusions could be improved.

The English (style, grammar and punctuation) could be improved; for example:

Line 37. The words “These sections are used to divide the article’s text” may be better stated as “The remainder of the paper is structured as follows”.

Lines 33- 36 could be improved and may be better stated as “Previous literature has employed graph reduction techniques to analyze process models [20] and utilised specialized tools, frameworks and methodologies for process analysis and modelling [22, 23. 24]”.

Line 77. The order of the various plans should match the order in which they are later discussed. The “Trauma plan” is also referred to as the “Traumatology plan”.

Lines 118-119. The words “we can mention” are not required.

Line 183. The word “paper” is not required. This also applies to other references mentioned in the text.

Line 187. The words “in general” are not required.

Line 200. The word “picture” should be “figure”.

Line 223. The words “She” or “her” should not be used. This also applies to other parts of the paper.

Line 227. The word “he” should not be used. This also applies to other parts of the paper.

Line 256. To whom does “We” refer?

Some of the references need further detail. See, for example, references 26, 27, 28, and 29.

Author Response

(The authors gave the same response as above.)

Round 2

Reviewer 1 Report

Dear Authors,

    Thank you for your valuable feedback. The article is a suitable and can be publicated. 

Author Response

Thank you very much for reviewing our article; your advice was invaluable in the first round of the review process. We believe that the reader will be pleased with the publication of this text.

Reviewer 2 Report

The Authors adhered to the guidance provided in the original review. 

The construction of the scientific article, however, still cannot be considered correct. In the "Introduction" section, the order of the presented goals (according to the later order in the research process) should be changed. But that's just a minor point.

As for the structure of the article:

  1. I still think that the "Discussion" part needs to be strengthened - there are no references to the literature presented earlier. The limitations of the research process should be moved to the "Conclusions" section, as well as the identification of future research areas.
  2. Thus, the "Conclusion" section should be reinforced with the above elements.

I believe that the Authors can still apply for the publication of the article, of course, after making the marked corrections.

Author Response

Thank you to the reviewer for reviewing the article and providing insightful comments. Point by point answers are included in the document.

This manuscript is a resubmission of an earlier submission. The following is a list of the peer review reports and author responses from that submission.